

**INVESTIGATING METAMODELING CAPABILITY TO PREDICT SEA**
**LEVELS AND MARINE FLOODING MAPS FOR EARLY-WARNING**
**SYSTEMS: APPLICATION ON THE ARCACHON LAGOON (FRANCE)**
Sophie Lecacheux[1], Jeremy Rohmer[1], Eva Membrado[1], Rodrigo Pedreros[1], Andrea Filippini[1],
Deborah Idier[1], Servane Gueben-Vénière[2], Denis Paradis[3], Alice Dalphinet[3], David Ayache[3]
[1]BRGM, 3 av. C. Guillemin - 45060 Orléans - France
[2]KEYROS, 3 bis rue Jules Vallès - 75011 Paris - France
[3]METEO-FRANCE, DIROP/MAR/DAS - Toulouse
*Correspondence to: S. Lecacheux (s.lecacheux@brgm.fr)*
**Abstract -** Marine flooding events during storms are expected to occur more frequently due to
sea level rise. Hence, early warning systems (EWS) dedicated to marine flooding are expected
to develop in the coming years. In this study, we compare three data-driven methodologies to
overcome the computational burden of numerical simulations. They are all based on the
statistical analysis of pre-calculated databases, to downscale total sea levels and to predict
marine flooding maps from offshore metocean operational forecasts. While the first one is a
simple analog-based research from offshore metocean conditions, the next two both use a
machine learning type metamodel to predict total sea levels at the coast, and either an analog
or a deep-learning approach to predict marine flooding maps. The analysis, carried out with a
cross-validation exercise and historical storms on the pilot site of Arcachon lagoon (Southwest
of France), reveals that the analog-based approach is a valuable first step to explore the dataset
and improve the understanding of flooding phenomena, but lack precision for operational
forecast applications. On the other hand, the two metamodel-based approaches are more
suitable for fast prediction with a lower prediction error of inland water heights for the deep-
learning approach (on the order of 10 cm). Both approaches can then be complementary
depending on the type of event, the required level of prediction accuracy to support operational
decision making, and the forecast lead time. In this sense, the study also underlines the
usefulness of precalculated databases to conduct a preparatory work with crisis managers to
determine the type of information and the right level of complexity required to address
operational needs.
**Keywords:** Marine flooding, maps, forecast, metamodels, Arcachon lagoon



## 1 Introduction

The high rates of population growth and urbanization in coastal regions tend to constantly increase marine flooding risks in low-lying areas. Projections estimate the number of people living in coastal flood-prone areas at more than 200 million by 2100 (Hauer et al., 2021). Marine flooding is a phenomenon resulting from the combination between various processes generated at different time and space scales (atmospheric circulation, waves, atmospheric surge, tide, and sometimes river discharge) and the local configuration of the coast (coastal bathymetry and elevation, protection structures, land use, hydraulic networks, etc.). The numerous variables, scales and sources of uncertainty make marine flooding events very complex to predict several hours or days in advance. However, marine conditions forecasts (Toledano et al., 2022; Lorente et al., 2019) and coastal flood early warning systems (Stansby et al., 2012; Dietrich et al., 2013; Apecechea et al., 2023) have gained a significant impulse in the last decades. Today, there is a growing demand not only to shift towards ever more local marine flooding forecasts, but also to better account for uncertainties, which requires approaches using multi-sources or even probabilistic metocean conditions (see a recent discussion by Turner et al., 2024).

Recent improvements in high performance computing enabled numerical weather prediction systems to move from deterministic to probabilistic forecasting using Ensemble Prediction Systems (EPS) (Descamps et al., 2015). While EPS are increasingly used to predict river flows and induced continental floods in several countries (Wu et al., 2020), it is only emerging for marine flooding (Hawkes et al., 2008; Lecacheux et al., 2018: Biolchi et al., 2022). Despite ongoing efforts to develop new generations of high performance oceanographic and phase-resolving models accounting for complex processes (Filippini et al., 2018), the main challenge still remains the computer power required to run multiple simulations with a chain of models of increasing resolutions (from a hundred meters for coastal waves and surges to a few meters for marine flooding). Most of the examples of – deterministic or probabilistic – systems forecasting explicitly marine flooding rely on three solutions to overcome these limitations: High Performance Computing, reduced process complexity with generalized overflowing models or empirical formula, statistical analysis of pre-calculated flooding scenarios (Lecacheux et al., 2021; Beuzen et al., 2019). Now, operational needs and constraints of potential users must guide these technical choices. It is about linking top-down and bottom-up approaches by connecting technical capabilities with local users' needs to finally identify the



suitable level of complexity able to produce the right level of information (parameters,
resolution, and precision) for decision making (Demerrit et al., 2016; Tarchiani et al., 2020).
In this study, we focus on methodologies based on pre-calculated databases to downscale total
sea levels and marine flooding maps from offshore metocean conditions as developed at
European scale in ECFAS project (Le Gal et al, 2023 and 2024). If databases are often used to
work by analogy, based on storm similarity with forecasted conditions, this solution also
provides complementary possibilities to (1) improve the understanding of marine flooding
processes and identify key scenarios and action plans to facilitate foresight and anticipation in
crisis management (Luesink et al., 2024) (2) apply machine learning (denoted ML) type
methods to develop very rapid forecasting models (or metamodels) able to replace numerical
models to carry out simulations in real time. ML-based metamodeling techniques have made
great progresses in this field of application (hurricane: Torres et al., 2020; estuarine: Parket et
al., 2019; port: Bolle et al., 2018; tsunami: Blaser et al., 2019, Denamiel et al., 2019) and opens
up encouraging perspectives for marine flooding forecasting. Through a statistical analysis of
pre-calculated training databases, it can predict key flooding indicators (surge, discharge, water
height, etc.) at a given spatial location of interest within reasonable time and computing
resources while preserving the accuracy of full process models. Yet, some issues remain to push
this metamodel-based approach toward operational applications, and more specifically the
production of spatialized indicators such as inland water height maps, which is still a matter of
active research (e.g., Perrin et al., 2020; Lopez-Lopera et al., 2020). Indeed, producing such
spatialized indicators requires to overcome the difficulty in learning complex mathematical
relationships between the metocean forcing conditions and flood maps (i.e. output of high
dimensional related to the number of pixels, typically of several 10,000s).
The objective is then to investigate different approaches, based on pre-calculated databases,
to produce information that is relevant for operational forecast applications. This implies
analysing not only the added value in terms of accuracy regarding the required implementation
effort but also in terms of adequacy with needs of crisis managers. We consider three
approaches with increasing complexity. The first one is a simple analog-based method, which
is based on the analysis of the similarity between the coming metocean conditions and an
individual event in the pre-calculated training database (also named as "look-up table"). The
next two are ML-based methods aiming to supplement or substitute the analog-based approach
with (1) a regression-type metamodel to predict total sea levels in the lagoon; (2) deep learning



techniques to predict the marine flooding maps. To support the analysis, we focus on the
Arcachon lagoon (Southwest of France, Fig. 1) where a database of a few hundred flood
scenarios have been simulated for the Service of Civil Defense and a participatory work with
crisis managers enabled to identify operational needs and critical thresholds associated with
graduated categories of flood impacts.

6       In the following, we present the pilot site, the databases and lessons learnt from their analysis

with crisis managers in Sect. 2, the three approaches for sea levels and marine flooding
prediction in Sect. 3, and the comparison of their skills on in Sect. 4. In the discussion in Sect.
5, we analyze the pros and cons of the different methods regarding local users' needs while
Sect. 6 summarizes the main conclusions and addresses the perspectives for application of these
models for operational forecast.
**2    Site, databases setup and analysis**
**2.1    Site description**

14       Arcachon lagoon is a large low-lying area located in a semi-closed environment and mainly

subject to marine overflows driven by offshore storm surge and wave forcing associated with
local winds (Fig. 1, A). It is a well-studied site, which benefits from a rich bibliography,
covering both its general functioning (Bouchet et al, 1997), the geological and
morphosedimentary context (Saltel et al., 2014), the hydrological functioning (IFREMER,
1997) and oceanographic characteristics (Castelle et al., 2017; Dupuis et al., 2016). In recent
years the lagoon has been regularly affected by minor to moderate marine flooding notably
during storms Klaus (2009), Xynthia (2010), and more recently Céline (2023). As illustrated
on Fig. 1 (B), these storms generated sea levels (tide + storm surge) from annual (~3 m NGF)
to approximately 10-years (~3.5 m NGF) return periods at the Arcachon-Eyrac tide gauge but
there is no historical example of event beyond. Yet, statistical and hazard studies (Mugica et
al., 2014; SHOM, 2022) demonstrated that a 100-year return period event could reach up to 3.7
NGF at the same location, then affecting almost ten thousand of persons (SIBA, 2015).





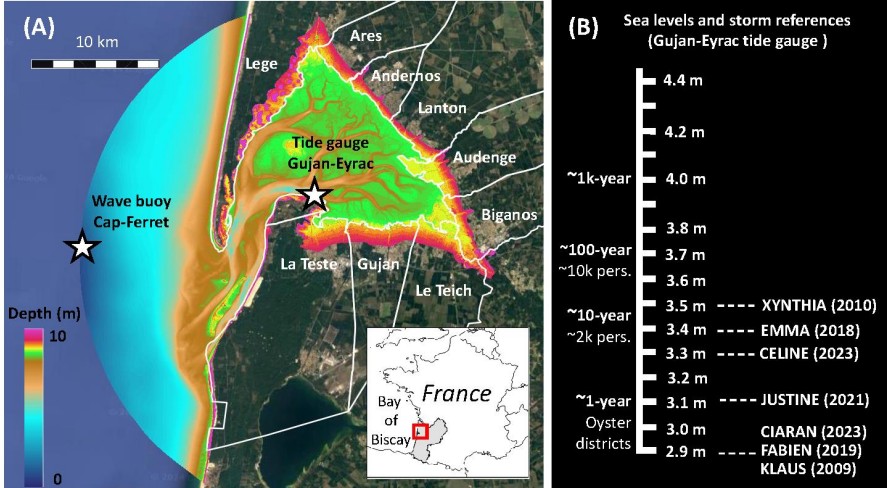

**Figure 1: A) Location and bathymetric map of the study site: white stars indicate the location of the tide gauge named Arcachon-Eyrac (sea level) and the wave buoy named Cap-Ferret. B) Sea levels (m NGF) with associated return periods and historical storms references. Background is from © Google Earth**

Based on this bibliography and historical events, the main elements to consider when modeling coastal floods on this area are:

- Marine flooding by overflowing is dominant on the scale of the entire lagoon even if local sectors can also be exposed to wave overtopping (as the seafront of Andernos, see location in Fig. 1, A).

- Waves have a significant effect on marine flooding through the generation of a homogeneous wave setup inside the lagoon (up to several tens of centimeters) generated by the breaking of the swell at entrance. of the lagoon).

- West/northwest winds are the most frequent and can generate an additional storm surge inside the lagoon particularly at the extreme east (from Biganos to Arès) where sea levels may reach several tens of centimeters higher than in Arcachon in case of strong winds;

- Although river floods are characteristic hazards of wetlands such as the Arcachon lagoon, available studies and measures does not enable to characterize the conjunction between storm surge and river floods on this site (in particular in the Leyre delta between Le Teich and Biganos).

In this study we focus on marine overflowing only, without considering neither wave overtopping (localized on the seafront of Andernos) nor the conjunctions of river flood and storm surge (located in the Leyre Delta).



**2.2    Design of scenarios for learning and validation of the metamodels**
The synthetic storm conditions constituting the scenarios of the pre-calculated database
(used for the analysis of the physical processes and for setting up the ML-based techniques, see
Sect. 3) are built on the basis of a tri-variate statistical analysis of the extremes of wind, wave
and surge conditions at the Cap-Ferret wave buoy located at a depth of 50 m in front of the
lagoon (Fig. 2, A). The reference data are extracted at high tide from reanalyzes over the period
1979-2009: CFSR (Saha et al., 2010) for wind, ANEMOC 3 (Raoult et al., 2018) for waves and
MARS30 (Mugica et al., 2014) for the storm surge. After fitting probability laws for significant
wave heights (Hs), skew surge (SPM) and wind intensity (U) using the Generalized Pareto law
(Coles et al., 2001), the dependence models are adjusted for U, Hs and SPM according to the
conditional extreme model of Hefferman and Tawn (2004) while managing the co-variables
(peak period Tp, wave directions Dp and winds Du). Finally, many random combinations (here
chosen at 150,000) with the same statistical characteristics as the input data are generated via
Monte Carlo simulations. To constitute the learning database, 50 moderate to strong storm
conditions (Hs>4m) are selected via a clustering algorithm (Camus et al., 2011) aimed at
maximizing the diversity of scenarios (Fig. 2, A). These 50 conditions are associated with 10
tide levels between coefficients 40 and 120 (i.e. varying from 1 m to 2.8 m NGF at the tide
gauge every 20 cm), bringing the total number of scenarios to 500. Each scenario is therefore
described by 6 storm parameters at the buoy (U, Du, Hs, Tp, Dp, SPM) and a high tide level
(T) at the gauge.
These synthetic events are completed with historical and pseudo-historical events for
validation and demonstration purposes. Historical events (Table 1) correspond to 8 storms
(Klaus, Xynthia, Emma, Fabien, Justine, Ciaran, Domingos, Céline) that occurred between
2009 and 2023 and whose forcing conditions are extracted from observed winds and waves as
well as reanalyzed storm surge reanalyzes at the Cap-Ferret wave buoy location. Table 1 and
Fig. 2 (A) shows their characteristics at high tide and Fig. 2 (B) an example of parameters time
series for storm Xynthia. Although they are well scattered in the synthetic dataset range, these
events remained quite limited in terms of total sea level (Fig. 1, B) and then marine flooding
impact (as mentioned in Sect. 2.1) because the most intense storms (Fabien, Klaus and
Domingos) occurred at low tide coefficients. Thus, "pseudo-historical" events are added by
combining these 8 storms with the same high tide levels as the synthetic events, to create marine





flooding events of different severity (greater or lower than historical events). This results in 80
additional events hereafter named Pseudo_*StormName*_T*ideLevel(NGF)*.

| | XYNTHIA | KLAUS | EMMA | JUSTINE | FABIEN | CELINE | CIARAN | DOMINGOS |
|---|---|---|---|---|---|---|---|---|
| SPM (m) | 0.44 | 0.49 | 0.33 | 0.21 | 0.38 | 0.27 | 0.3 | 0.47 |
| Hs (m) | 6.7 | 7 | 6.46 | 8 | 8.57 | 4.25 | 7.36 | 8.78 |
| Tp (s) | 12 | 14 | 11 | 16 | 13.5 | 14 | 13 | 13.5 |
| Dp (°) | 266 | 277 | 251 | 273 | 276 | 290 | 280 | 275 |
| U (m.s$^{-1}$) | 23.5 | 28 | 23.5 | 19.5 | 28 | 11.5 | 20 | 25 |
| Du (°) | 250 | 230 | 250 | 280 | 270 | 190 | 270 | 250 |
| Tide (m) | 2.74 | 1.73 | 2.71 | 2.46 | 1.85 | 2.77 | 2.07 | 1.3 |

**Table 1:  Storm parameters (U, Du, Hs, Tp, Dp, SPM) at high tide for the 8 historical storms.**

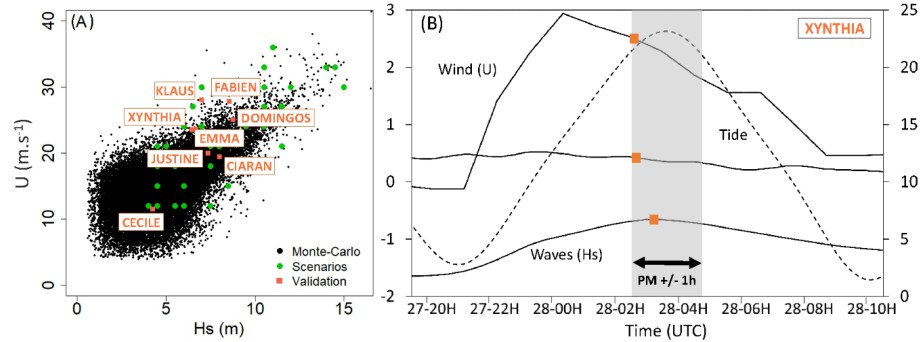

**Figure 2: A) Monte-Carlo simulations (black points), synthetic (green points) and historical or validation**
**(orange points) storm conditions at high tide. B) Example of storm parameters time series and extraction**
**of conditions at high tide (orange points) for storm Xynthia (2010).**
**2.3    Flood numerical model set up and simulation of the scenarios**
The marine flooding model includes a chaining of WW3 wave models (Tolman, 2014) and
UHAINA hydrodynamic models (Filippini et al., 2024) on an unstructured mesh with a
resolution from kilometric offshore (at 50 m depth) to decametric on land (Fig. 3). It can
simulate the propagation of waves and sea levels inside the lagoon, the generation and
propagation of wave setup, the additional surge induced by local winds, and the marine flooding
by overflowing.  Despite the resolution of the mesh remains limited on land, the model
represents the main obstacles for water flows thanks to the line constraints representing the
protection structures (dikes) but also little walls, embankments and roads. For each point of the
grid, an altitude is extracted from a digital terrain model (DTM) built with a combination of
bathymetric surveys with resolutions of 100 m offshore and 10 m inside the lagoon as well as



LIDAR data from 2016 survey on land. Finally, the land cover is represented by a manning
coefficient extracted from friction maps (Mugica et al., 2014).

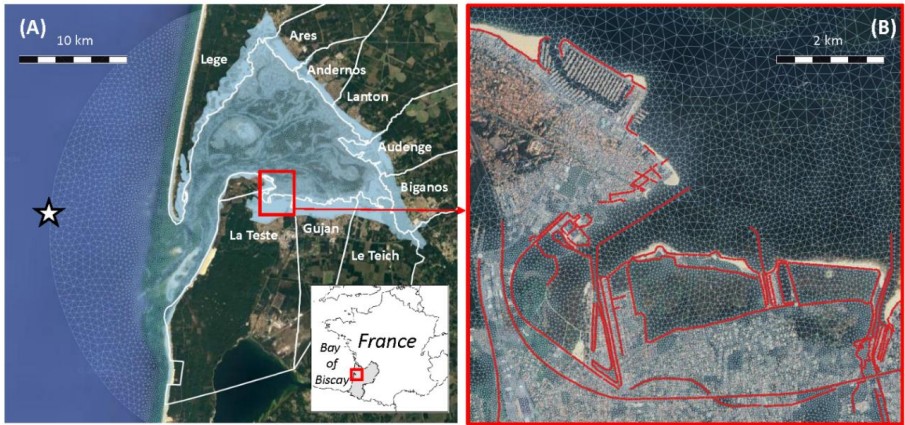

**Figure 3: A) Unstructured mesh used in the modelling chain (UHAINA and WW3) at the scale of the**
**lagoon. B) Zoom centred on La-Teste-de-Buch. Background is from © Google Earth**
A fundamental difference between synthetic cases (for learning of metamodels) and
historical or pseudo-historical cases (for validation of metamodels) is the consideration of
temporality. While synthetic cases are only described by a set of scalars representing the storm
conditions at high tide (U, Du, Hs, Tp, Dp, SPM), historical and pseudo-historical cases are
described with time series of the same parameters, from which conditions at high tide can be
extracted, but temporal variations are also available. To evaluate the influence of the temporal
variations of these conditions before and after high tide, sensitivity tests are carried out by
simulating the 8 historical storms of the validation set on a tidal cycle with both dynamic
conditions (i.e. with time varying time series) and stationary conditions (i.e. with the conditions
extracted at high tide (+/- 1h). For the maximum sea level over time (SSHmax), the comparisons
between the 8 storm simulations (dynamic and stationary) and tide gauge observations show
errors less than 10 cm (Table 2). Concerning flooded surfaces and maximum water height on
land over time (Hmax), the flooded sectors are also generally well reproduced and very similar
for dynamic and stationary simulations. Figure 4 presents an example for Xynthia (in Gujan-
Mestras and Le Teich) and shows that dynamic and stationary simulations result in very similar
patterns as observations despite SSHmax differences of 8 cm. The interested reader will find
further information about simulations sensitivity tests in Lecacheux et al. (2013) which
corroborate that (1) the modeling chain reproduces correctly SSHmax and Hmax on the ten



municipalities (2) SSHmax and Hmax are mainly controlled by conditions at high tide, which
justifies the use of storm parameters at high tide (U, Du, Hs, Tp, Dp, SPM) in the simulations.

|  | XYNTHIA | KLAUS | EMMA | JUSTINE | FABIEN | CELINE | CIARAN | DOMINGOS |
|---|---|---|---|---|---|---|---|---|
| **OBS** | 3.58 | 2.92 | 3.38 | 3.19 | 2.92 | 3.37 | 2.86 | 2.46 |
| **DYN** | 3.51 | 2.9 | 3.4 | 3.2 | 2.85 | 3.23 | 2.82 | 2.36 |
| **STAT** | 3.59 | 2.86 | 3.41 | 3.2 | 2.9 | 3.24 | 2.84 | 2.39 |

**Table 2: Comparison between SSHmax (in meters) at Arcachon tide gauge for the 8 historical events**
**observed (OBS), modelled with dynamic (DYN) or stationary simulations (STAT).**

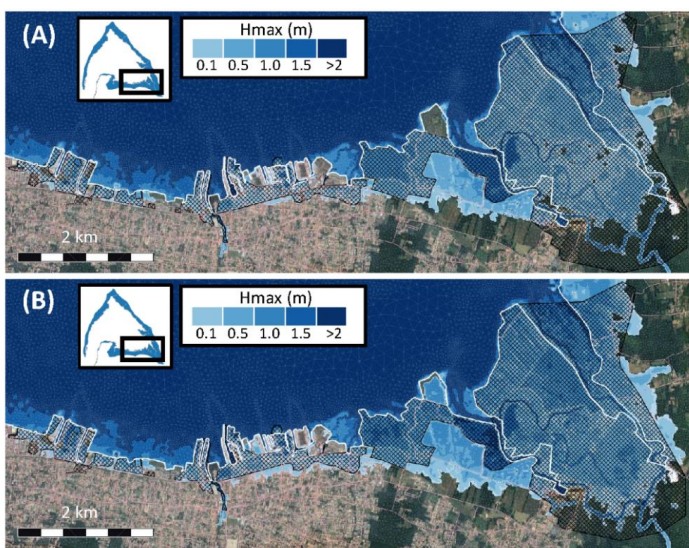

**Figure 4: Comparison between observed and dynamic (A) and stationary (B) simulations of flooded areas**
**for Xynthia in Gujan Mestras and Le Teich. Background is from © Google Earth**
Scenarios of the learning and validation datasets are thus simulated on a tidal cycle by
applying stationary conditions of surge, wind and waves on the entire tidal cycle. For each
event, simulations are carried out sequentially, starting from the highest high tide level to the
lowest, and stopping at the last overflowing level (corresponding approximately to SSHmax of
3 m NGF inside the lagoon). It induces that strong storm events are simulated with more tide
levels than moderate ones that generate marine flooding only for the highest tidal coefficients.
To sum-up, the learning dataset is composed of 220 simulated scenario and the validation
dataset of 32 simulated scenarios all described by (1) forcing conditions of 6 storm parameters
at the buoy (U, Du, Hs, Tp, Dp, SPM) and a high tide level (T) at the gauge (2) maps of
maximum sea level in the lagoon (SSHmax) and water levels on land (Hmax).




## 2.4    Overview and analysis of the datasets with crisis managers

SSHmax obtained with the simulations of learning scenarios vary from 2.6 m to 4.4 m NGF at the tide gauge. For comparison, the annual and centennial levels at the same location are respectively 3 m and 3.7 m NGF (Fig. 1). The learning dataset thus enables to describe a wide range of events from annual to exceptional (more than millennial).

The exposure of the municipalities surrounding the lagoon is although quite variable. Figure 5 presents two maps showing the percentage of scenarios generating marine flooding (A) as well as SSHmax and Hmax induced by the most intense scenario (B). They highlight that:

- Municipalities with wetlands, either limited (La-Teste-de-Buch, Lanton and Ares) or quite extended (Le Teich, Biganos, Audenge) present a greater general exposure with a higher rate of flooding and higher potential Hmax (although it concerns non-urbanized areas).
- Municipalities without wetlands (Arcachon, Gujan-Mestras, Andernos, Lege-Cap-Ferret) are mainly exposed near the seafront with a rapid decrease of flood rate and potential Hmax when moving further inland.

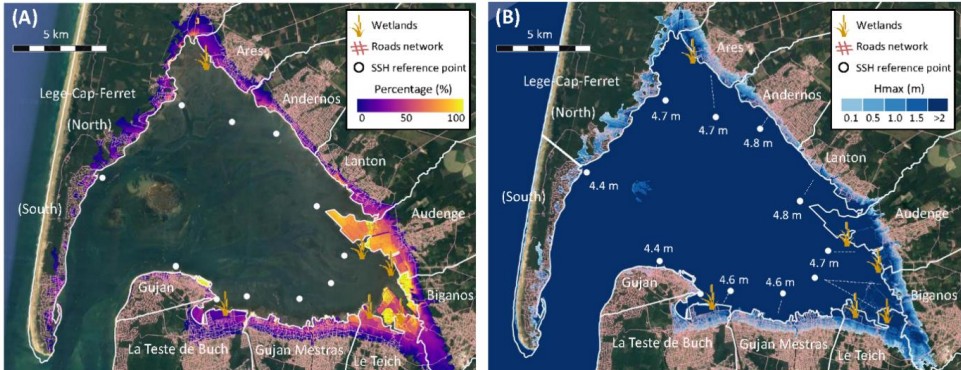

**Figure 5: A) Percentage of scenarios (calculated from the learning database) leading to marine flooding on each node of the model mesh. B) Sea levels denoted SSHmax on the municipalities reference points and Marine flooding map (Hmax) for the most intense scenario. Background is from © Google Earth**

But being exposed does not necessary mean being vulnerable. The impact of a flooding event depends on its severity but also on the stakes involved and the problematics encountered by crisis managers. Crisis managers need aggregated indicators to strengthen their capacity to understand the information and take actions. To address this issue on the Arcachon lagoon, a collaborative work was realized with the municipalities to classify the scenarios (between 1-





year and 1000-years return period) into 3 categories (CAT) representing graduated levels of
impact and related to specific action plans:
- CAT 1 corresponds to limited flooding events concerning only seafronts, port docks or
oyster farming districts that are regularly subject to marine flooding and for which the
municipalities are well prepared and autonomous to manage the events.
- CAT 2 corresponds to moderate flooding events concerning few residential areas, transport
or energy networks, and for which logistical capacity of the municipalities is not exceeded.
- CAT 3 finally corresponds to major events (either by their extension or because critical
infrastructures are affected) for which the municipalities need help to manage the events.
As marine flooding patterns are mainly controlled by local SSHmax, these 3 main categories
(CAT 1, CAT 2, CAT 3) have been translated into thresholds of SSHmax on 10 reference points
in front of each municipality (Fig. 5, B). Finally, they correspond to ranges of SSHmax of about
30 cm, but some of them can reach up to 50 cm. The repartition of the categories for each
municipality and the position of the scenarios of the learning database (box plots) and validation
database (points) showed on Fig. 6 enables to say that:
- Even if the municipalities comprising wetlands (notably Biganos, Audenge, Ares: see Fig.
1) are more exposed to marine flooding, urban areas and critical stakes are really impacted
(with events of CAT 2 or 3) for higher SSHmax than the other municipalities.
- The learning scenarios (box plots) are rather well distributed across the categories.
- Historical events do not exceed CAT 2 (with Xynthia) but pseudo-historical events complete
the validation scenario relatively well to represent all the categories.

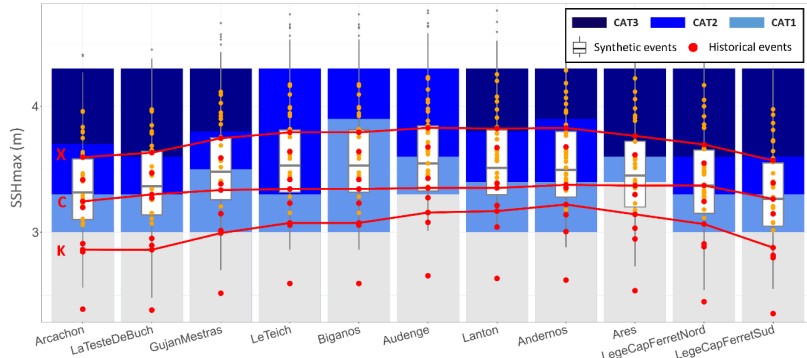

**Figure 6: Categories of impact (CAT) related to SSHmax and repartition of the scenarios of the datasets:**
**synthetic (box plots), historical (red points) and pseudo-historical (orange points). Red lines illustrate the**
**variations of SSHmax for Klaus (K), Céline (C) and Xynthia (X).**





This preliminary work highlighted the necessity to forecast sea levels (SSHmax) and the
associated categories of impact (CAT) for each municipality first, so that crisis managers can
refer to typologies of action plans, the interest in forecasting marine flooding maps (Hmax)
coming in a second step to complete and refine the description of the event within the category.
In the following, we thus propose different methods enabling to articulate the prediction of both
SSHmax and Hmax with different levels of accuracy.
**3    (Meta)modeling methods**
In this section, we describe the formal framework for setting up the ML-based models
(named metamodels). Let us consider the maximum value of water height induced by flooding
at the N spatial locations denoted $H_{\max}^i$ defined by their geographical coordinates $\mathbf{s}=(s_1; s_2)$. Let
us define $\boldsymbol{H}_{\max}^i = \left(H_{\max}^i(\mathbf{s}_1), H_{\max}^i(\mathbf{s}_2), \ldots, H_{\max}^i(\mathbf{s}_N)\right)$ the vector of Hmax related to the i-th
model run defined by the vector of $d$ offshore conditions $\boldsymbol{x}^i = \left(x_1^i, x_2^i, \ldots, x_d^i\right)$, and the n×N
matrix $\boldsymbol{H}_{\max} = \left[\boldsymbol{H}_{\max}^1, \boldsymbol{H}_{\max}^2, \ldots, \boldsymbol{H}_{\max}^n\right]'$ built from the n numerical model's outputs. The
objective is to predict $\boldsymbol{H}_{\max}$ given $\mathbf{x}$ based on a statistical predictive model (named metamodel)
that is trained with the learning dataset.

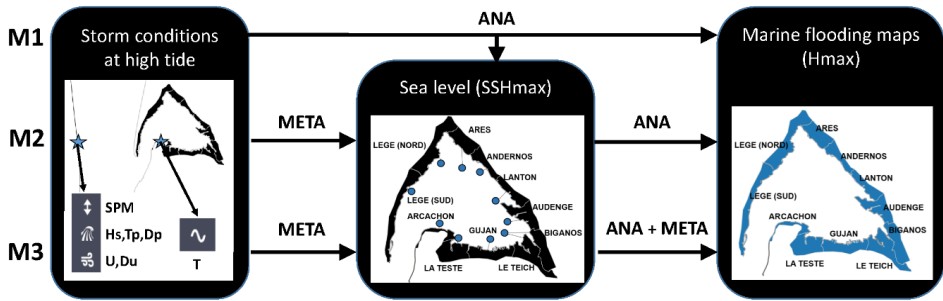

Figure 7: Diagram of the 3 methods named M1, M2 and M3 that either rely on an analog-based
approach (named ANA) or on a combination with metamodeling methods (named META)
**3.1    Description**
**3.1.1    M1: Analog-based approach**
The first one, named "Analog", is based on a look-up table approach. For given metocean
forcing conditions, it consists in querying within the learning database SSHmax and Hmax
maps for the scenario whose conditions that are the most similar. The result is named "analog"



in reference to the approach commonly used in weather forecasting (e.g. Van den Dool, 2007).
To measure the similarity, we use the distance named "EC" proposed by Camus et al. (2011)
to both handle continuous and circular data (wind and wave direction). In this study, we only
query one single analog, because our preliminary tests using multiple analogs (for instance the
three analogs whose forcing conditions are the first three closest to the targeted one) have shown
minor increase of the predictive capability of this approach.

### 3.1.2   M2: Combined Gp metamodel and analog

In this approach, we first predict the sea levels SSHmax from the metocean conditions at
the reference points (outlined in white dots in Fig. 5) with a Gaussian process (denoted Gp)
regression model (Williams and Rasmussen, 2006). On this basis, we identify for each
municipality the analog's flood map in the learning database that has the closest SSHmax to the
Gp-based predicted one. The choice of the Gp model is guided by the high performance of this
method as shown by multiple real case studies in coastal flooding (see e.g., Rohmer and Idier,
2012; Jia and Taflanidis, 2013; Perrin et al., 2020; Lopez-Lopera et al., 2020). In the following,
we assume a Gp model with a linear trend, a Matérn 5/2 kernel model and a nugget effect. The
hyper-parameters (parameters of the Gp model) are all evaluated through maximum likelihood
estimation (e.g. Roustant et al., 2012). See further technical details in Appendix A.

### 3.1.3   M3: Combined Gp metamodel and dimension reduction autoencoder

The third approach improves the second approach by predicting Hmax from the forcing
condition and SSHmax with a prediction based on the combined dimension reduction (denoted
DR) – Gp metamodel approach that has been proposed in the literature (see for instance Jia et
al. (2013) for hurricane-induced waves, Ma et al. (2022) for hurricane-induced surges, Li et al.
(2020) in an estuary context, Perrin et al. (2020) for storm induced coastal flooding, etc.). To
overcome the high dimensionality of the flood map (related to the number of pixels typically
of several 10,000s), the DR technique aims to extract a much smaller number (typically less
than 10) of new variables (named latent) to represent the very high-dimensional output.
Formally this consists in transforming $\boldsymbol{H}_{\max}$ into a finite number of latent output variables
$\mathbf{Z} = [\mathbf{z}^1, \mathbf{z}^2, \dots, \mathbf{z}^n]'$ with $\mathbf{z}^i = \left(z_1^i, z_2^i, \dots, z_T^i\right)$ so that $T\!<\!<\!N$.
A popular approach is based on principal component analysis, denoted PCA (Jolliffe 2002).
However, PCA is a linear technique, and it has been shown in our case to have limits (Rohmer



et al., 2024a) due to the complexity of the flood maps, which have discontinuous heterogeneous
patterns. Therefore, we preferably rely on a non-linear DR, namely a neural-network-based
autoencoder, denoted AE (Baldi, 2012), in place of the popular PCA. For each of the latent
variables, a Gp model is trained to link the latent variables to SSHmax forcing conditions.
Recently Rohmer et al. (2023) have shown the added value of accounting for the dependence
between the latent variables. To do so, we rely on the multi-output Gp model proposed by Gu
and Berger (2016) (named the parallel partial kriging model). The AE architecture (number of
nodes, number of layers, activation function) is selected so that the validation error is minimized
as for Rohmer et al. (2024a) (see Appendix B for the comparison of architectures of
autoencoders and Sect. 3.2 for the description of the validation procedure). Note that, if we
recalculate Hmax in the flooded area obtained with M2 with the combined dimension reduction
(denoted DR) – Gp metamodel, we keep the flood extent from M2 to compensate the tendency
of the chosen statistical approach to overestimate the flood extent (see Rohmer et al., 2024a).
**3.2    Procedure for performance assessment**
The objective is to measure to which extent the different approaches are respectively able to
correctly predict SSHmax on the reference points and whole maps of Hmax given "yet-unseen"
offshore metocean conditions. Two approaches are used:
(1)      The first approach relies on the available learning scenarios using a 10-fold cross-
validation procedure (e.g., Hastie et al. 2009). This holds as follows: (i) the initial training
dataset of input forcing conditions is randomly split into 10 equal sub-sets; (ii) a sub-set is
removed from the initial set, and a new RF model is constructed using the remaining set; (iii)
the sub-set removed from the initial set constitutes the verification set and the differences
between the true and the estimated value can then be estimated. In this procedure, the initial
random split at step (i) may have some influence on the results. This can be minimized by
repeating the procedure given number of times (typically 10 times).
(2)      The second validation approach uses the set of independent samples that have not been
used for the training. Here, we use the validation scenarios composed of historical and pseudo-
historical events that are in the range of the learning database (we exclude storm Domingos that
occurred with a tidal coefficient below 40 but we keep all the Pseudo_Domingos_XX whose
tidal coefficients are inside the tidal range of the learning database).





The prediction error is measured with the absolute error defined as at each location $\mathbf{s}_{j=1,\ldots,N}$:

$$AE^i(\mathbf{s}_j) = \left| H_{max}{}^i(\mathbf{s}_j) - \widehat{H}_{max}{}^i(\mathbf{s}_j) \right|, \tag{Eq. 1}$$

with $H_{max}{}^i(\mathbf{s}_j)$ the true Hmax at the j$^{th}$ location $\mathbf{s}$ (j=1,…,N) given the i$^{th}$ input offshore
conditions (i=1,…,n), and $\hat{y}^i(\mathbf{s}_j)$ is the reconstructed Hmax map using the metamodel approach.
A spatially averaged error indicator for the i$^{th}$ case is then defined as:

$$MAE^i = 1/N \sum_{j=1}^{N} AE^i(\mathbf{s}_j), \tag{Eq. 2}$$

## 4   Results
### 4.1   Performance results from the cross-validation procedure
Figure 8 presents the results of the 10-fold cross validation on SSHmax and Hmax for the
three methods presented in Sect. 3. For SSHmax, the error corresponds to the difference
between numerical simulations and metamodel-based predictions for each event on the
reference points of each municipality (see Fig. 5). For Hmax, the error corresponds to the
Mean Absolute Error (or MAE, see Sect. 3.2 and Eq. (2)) calculated for each municipality for
the flooded nodes where Hmax > 0 in the numerical simulation (results of MAE for hmax=0
in the simulation are also provided in appendix C to compare the performances of prediction
of the flood extent).
Concerning SSHmax, the analog-based approach (M1) enables to estimate the simulation
results with a MAE of 10 cm in average, but for some cases it can reach up to 30 cm. On the
other hand, using metamodels (in M2 and thus M3) enables to get closer to the target value
within 5 cm most of the time and within 10 cm in all cases. When looking at Hmax, we notice
that the MAE decreases by introducing successively metamodel-based approaches to estimate
SSHmax and Hmax (from M1 to M3). With M3, the median of the MAE is below 20 cm
whereas its ranges between 15 cm and 40 cm depending on the municipality for M1 and M2.
This first performance analysis based on cross validation shows that the two metamodels
are correctly trained and improve the accuracy of SSHmax and Hmax predictions. However,
we can observe differences between the municipalities, with potentially higher MAE (up to
60 cm) on whose comprising wetlands (Le Teich, Biganos, Audenge, Ares). A first
explanation is the deeper Hmax expected on wetlands. To better understand the sources of
errors, a deeper analysis is carried out in the next section with the validation dataset.





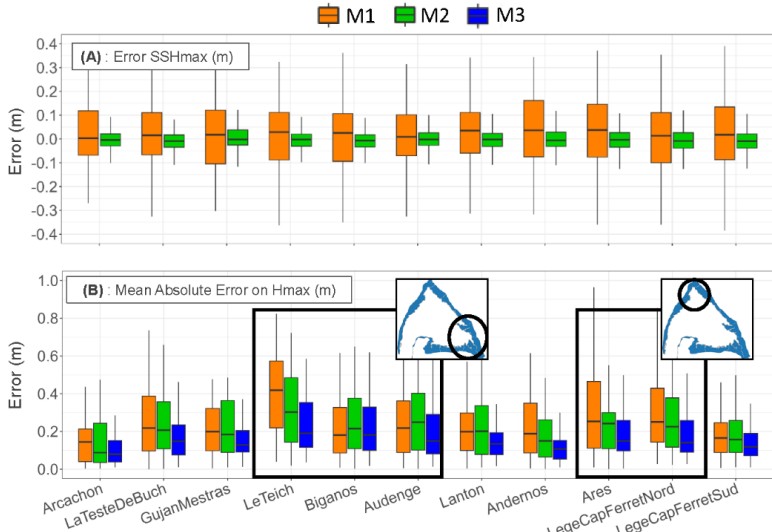

**Figure 8: Error on SSHmax (A) and Mean Absolute Errors for Hmax by restricting the analysis to the mesh nodes that are flooded, i.e. for Hmax > 0, in the numerical simulation (B) calculated with the 10-fold cross validation (Sect. 3.2). Box plots represent the 25th and 75th percentiles.**

## 4.2    Predictive performance on pseudo-historical cases

The analysis of the predictive performance with validation cases presented on Fig. 9 reveals the same ranges of errors than the cross validation with a global improvement of the prediction of SSHmax and Hmax from M1 to M3. However, some differences appear as predictions of SSHmax tend to be slightly overestimated for all the municipalities and M3 seems to perform better than in cross validation for municipalities with wetlands. These differences, due to the limited validation sample available, do not change the conclusions of Sect. 4.1.



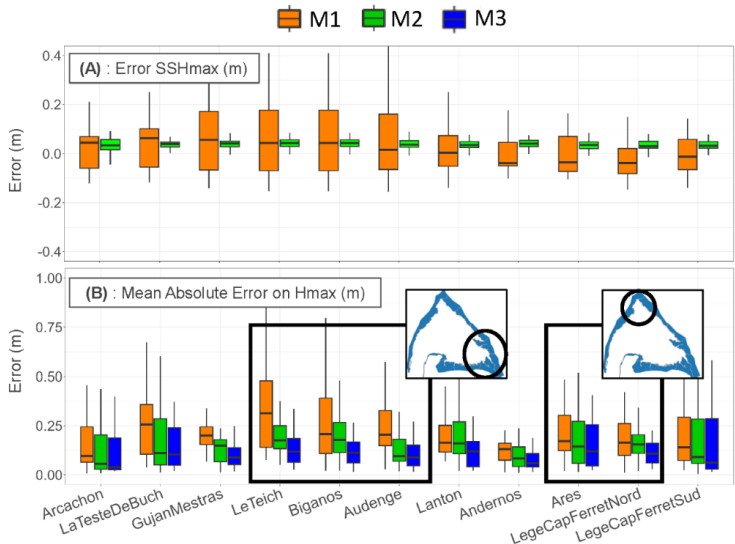

**Figure 9: Errors on SSHmax (A) and Mean Absolute Errors on Hmax by restricting the analysis to the mesh nodes that are flooded, i.e. for Hmax > 0, in the numerical simulation (B) calculated with the validation cases (Sect. 3.2). Box plots represent the 25th and 75th percentiles.**

If the MAE on Hmax enables to compare the prediction accuracy between the methods, it also aggregates different sources of errors depending on (1) the accuracy of SSHmax prediction (2) the marine flooding category (3) the exposure of the municipalities. To deepen this analysis, Figure 10 provides the correlation between errors on SSHmax and Hmax for each method (one point corresponds to the MAE of one municipality for one event). These plots show that:

- Higher errors on SSHmax logically lead to higher errors on Hmax but some scenarios among the most important errors on Hmax also present small errors on SSHmax.
- The maximum MAE on Hmax do not always correspond to major flooding events (of CAT 2 or 3 outlined in light blue and cyan).





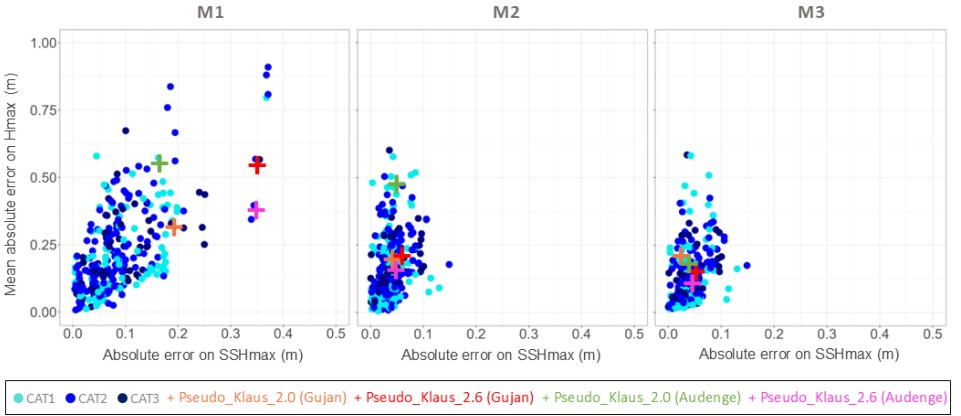

**Figure 10: Relation between SSHmax and Hmax errors for each method M1-M3. Each point represents the MAE calculated for one municipality and for one pseudo-historical event.**

To investigate further, we focus the analysis on two examples that illustrate different situations in Gujan Mestras and Audenge, namely Pseudo_Klaus_2.0, that is characterised by a low-level category CAT 1, and Pseudo_Klaus_2.6, that is characterised by a high-level category CAT2-3. For these events, MAE on Hmax for M2 and M3 remain quite high (between 20 cm and 50 cm) although SSHmax errors do not exceed 5 cm. The flooding and error maps plotted in Fig. 11 (for Pseudo_Klaus_2.0) and Fig. 12 (for Pseudo_Klaus_2.6) illustrate that, whatever the category of events, the maximum Hmax errors are localized in the seafront quarters (like in Gujan-Mestras) or inside wetlands (like in Audenge). These sectors both present (1) a higher sensitivity to threshold effects on SSHmax which has a direct effect on seafront flooding and controls the quantity of water discharged into low lying areas and (2) greater water heights and then potential Hmax prediction errors. Figures 10-12 also illustrate the improvement of marine flooding maps prediction from M1 to M3 both in terms of flooding extent and water height on land. Even for these cases with MAE of Hmax above the median, the flooding maps reproduced with M2 and more particularly M3 are similar to the flooding maps obtained with the simulation of the numerical model.



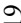

**Figure 11: Hmax and Hmax Absolute Errors (AE) for Pseudo_Klaus_2.0 at Gujan-Mestras (top) and Audenge (bottom) for the three methods.**

**Background is from © Google Earth**






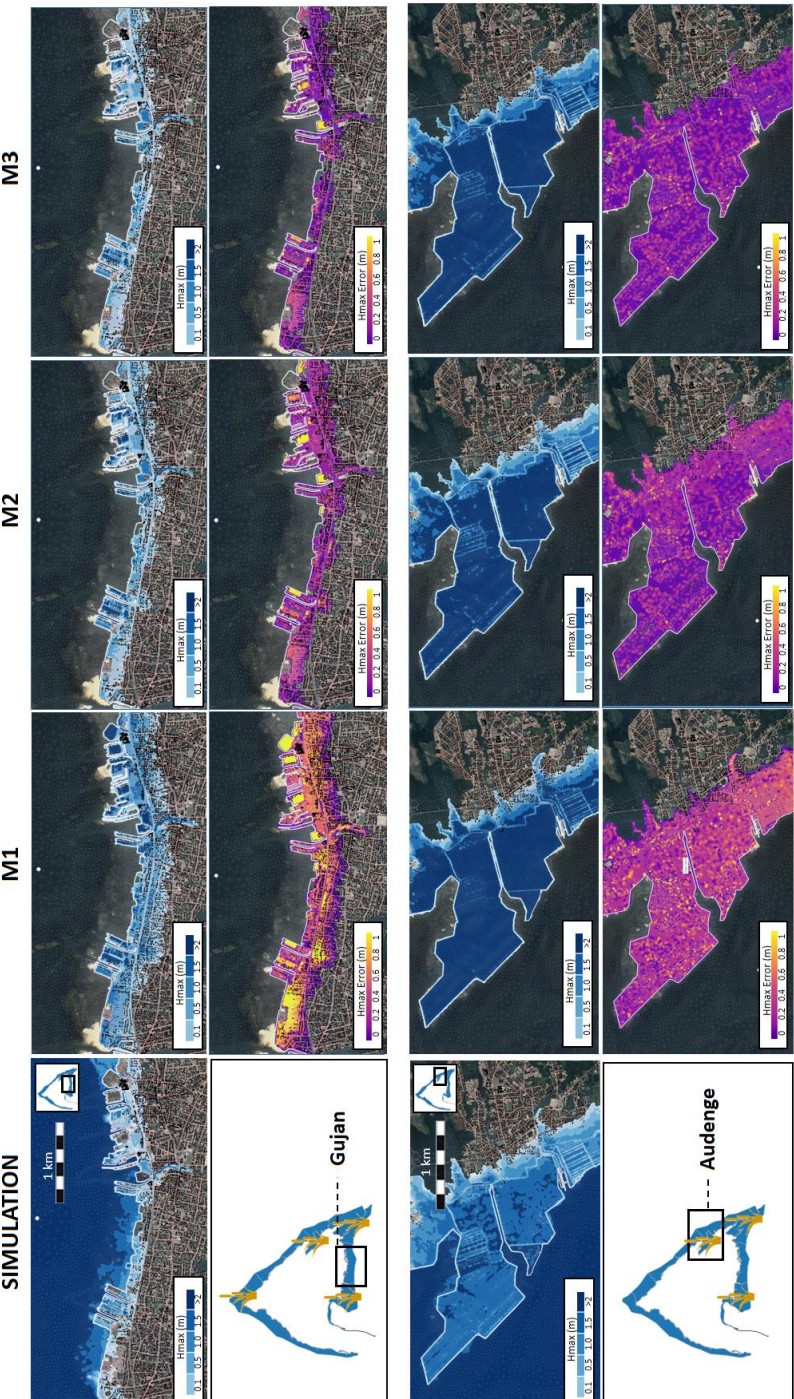

**Figure 12: Hmax and Hmax Absolute Errors (AE) for Pseudo_Klaus_2.6 on Gujan-Mestras (top) and Audenge (bottom) for the three methods. Background is from © Google Earth**

2



To complete the comparison of the performance between the three methods, we analyse the
confusion matrix on SSHmax and Hmax (Fig. 13) to check how often (all municipalities, mesh
nodes and validation scenarios combined) the right category of impact (through SSHmax) and
the right Hmax range (with classes of 50 cm commonly used for flood mapping) are predicted.
This shows that the categories of impact (CAT) are quite well predicted by M1 (more than 70%
of the test cases) but globally better predicted by M2 and M3 (more than 90% of the tests cases).
This result is in agreement with the errors on SSHmax presented in Fig. 9 (potentially up to 20-
30 cm for M1 and 10 cm for M2 and M3). Concerning Hmax range, M1 allows to predict the
correct order of magnitude only half of the time while M2 and M3 perform better with
satisfactory predictions respectively around 65% and 75% of the time. More generally, M2 and
M3 has a tendency of overestimation, but it should be underlined that the incorrect class is
almost always the closest, i.e. when misclassifying the '0.5-1' class, the predicted one is often
the '1-1.5'. From a risk perspective, this means that the approach minimises the false positives
but at the expense of false alarms.

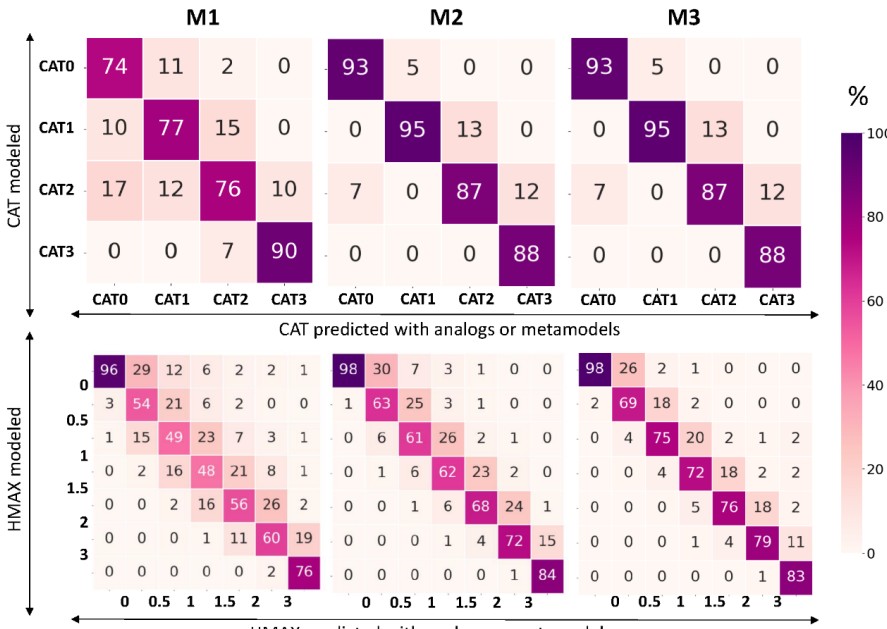

**Figure 13: Confusion matrix (or error matrix) for the impact categories denoted CAT (computed based**
17                                      **on SSHmax) (top) and Hmax (bottom).**



## 5   Discussion

Section 4 shows that the three methods can all be useful to predict maximum sea levels (SSHmax) as well as marine flooding category (CAT) and water height maps on land (Hmax), each of them bringing some advantages (see Tab. 3):

- The analog-based approach (M1), although very simple, can provide interesting first order estimates of SSHmax, CAT and Hmax but it can be limited and source of significant errors if the number of scenarios constituting the initial dataset is not enough to find sufficiently close analogs. Besides, the use of 6 input parameters to describe the storm conditions (U, Du, Hs, Tp, Dp, SPM) makes it difficult to evaluate if the selected analog overestimates or underestimates the target scenario, which can be delicate for operational forecast purposes.

- The approach combining Gp metamodels, analog and dimension reduction autoencoder methods (M3) is the most appropriate to jointly estimate with the highest accuracy, SSHmax (with a precision of 5 cm), CAT (90% of the cases) and Hmax (with a precision of 10 cm). It although requires more substantial implementation efforts, higher level of expertise, and uses statistical and machine learning methods that are still in the field of research for such forecasting purposes.

- Finally, the combined Gp metamodel and analog based approach (M2) turns out to be a good compromise between complexity/effort of implementation and accuracy. Through a quite simple statistical approach, it enables to estimate SSHmax and CAT with a good accuracy, which is an important first level of information for the crisis managers. Flooding maps are also quite well estimated, although less accurate than for M3 (with a higher prediction error for Hmax on the order of 20 cm).

To sum-up, the analog-based approach can be seen as a valuable first step to explore the dataset and improve the understanding of flooding phenomena. On the other hand, for an operational forecasting context, the two metamodel-based approaches are more suitable for fast prediction and can be complementary depending on the type of event and the forecast lead time. While M2 may reach sufficient level of accuracy to give first order estimates of categories of impact and marine flooding maps at 72h or 48h lead time (when uncertainties are still important), M3 may be more recommended at 24h or 12h lead time when a detailed vision of the flood map is necessary. As pointed out by the national flood hazard recommendations (MEDDL, 2014) and the municipalities involved in the study, the first 10 cm class of Hmax, is important for emergency services and crisis management as it determines the possibility to circulate easily





for pedestrians and vehicles. Thus, M3 (with a precision of about 10 cm for Hmax) is more
suitable to predict Hmax in areas with low Hmax and also minimize the errors on Hmax classes
of 50 cm range.

| | Method complexity | Precision | | | | Recommandation of use |
|---|---|---|---|---|---|---|
| | | SSHmax | CAT | Hmax (h>0) | Flood extent | |
| **M1** | + | 10 cm | 70% | 25 cm | + | Prevention and prepardness |
| **M2** | ++ | 5 cm | 90% | 20 cm | ++ | Forecast (few days lead time) |
| **M3** | +++ | 5 cm | 90% | 10 cm | +++ | Forecast (few hours lead time) |

**Table 3: Summary of prediction precisions and recommendations of use for the three methods. For**
**SSHmax and Hmax (h>0) the precision is an order of magnitude of the mean MAE for all municipalities**
**and all verification sets of cross validation. For CAT it corresponds to the percentage of correct**
**predictions (all categories mixed up). For the flood extent it is a qualitative appreciation of the quality of**
**the prediction based on Fig. 2B describing the errors for Hmax (h=0).**
Analysing the impact of uncertainties and the complementarity of different approaches for
operational forecast is worth being further analysed with concrete cases of storm forecasting.
This work, initiated in Rohmer et al. (2024b) will be continued and focused on applying these
metamodel-based approaches (M2 and M3) with forecasts from Météo-France deterministic
and EPS chains on recent events similarly as Dietrich et al. (2013) or Beuzen et al. (2019).
These experiments will enable to carry out a complete analysis of the relative sources of
uncertainties and to assess to which extent the metamodel error affects the spread of the
probabilistic forecast, and whether it can be neglected compared to the variability in the
metocean conditions.
Several lines for further improvements of the metamodeling methods are also identified.
First, we focused in this study on the cases that led to flooding for the construction of the
metamodels. Integrating also the case without flooding deserves to be investigated by testing
different approaches either by completing the approach with a classification step (see e.g.
Rohmer et al. (2018) for an example using a random forest classification technique) or using
variants of Gaussian process metamodels adapted to zero-censored data (Spiller et al., 2023).
More generally, this problem is related to the tendency of the metamodels to over-estimate the
flood spatial extent, and more advanced approaches should be tested to improve this aspect by



relying on more sophisticated deep learning techniques for instance based on generative models
(e.g. Ma et al., 2024) combined with appropriate optimisation approaches to select the most
optimal values of the ML model's parameters (called hyperparameters, see Bischl et al., 2023).

4       Finally, the global approach based on pre-calculated database is replicable on any type of

marine flooding-prone area, and whatever the physical processes at stake, as it manages
computation time issues by replacing real time numerical modelling by fast statistical tools.
The availability of the pre-calculated dataset enables to carry out an interesting preparatory
work with potential users from municipalities or state services in preparedness phase to better
assess the accurate thresholds, and the type and resolution of information required. This
preparatory work is essential to allow the users to determine the criteria that will guide the
whole developments and define the right level of complexity required to address operational
needs.
**6    Conclusion**
In this study, we developed and compared three methods to predict marine flooding maps
from offshore metocean conditions using successively analog-based approaches, regression
type metamodels and deep learning techniques. The comparison, based on both technical
accuracy and suitability for operational needs criteria, showed that the three methods all bring
some advantages and drawbacks. If the efforts required to develop full metamodel approaches
(including deep learning techniques) may be relevant for very urbanized sectors with civil
security issues as the pilot site of Arcachon lagoon, the combination of more simple regression
type models combined with analogs may be sufficient for applications at large scale or for
natural or agricultural areas (where lower precision may be acceptable).
In this way, this work offers encouraging perspectives on the use of pre-calculated databases
to carry out operational forecasts of marine flooding maps at local scale for different types of
sites and geographical coverage. In particular, the capacity to predict quickly marine flooding
maps opens up the possibility to address the issue of uncertainties though the production of
ensemble forecasts and in-depth sensitivity analyses.



**Appendix A: Kriging metamodelling**
For a given t=1,…,T, each n-dimensional vector $z_t = (z_t^1, z_t^2, …, z_t^n)$ is assumed, in the context
of kriging modelling (denoted KM), to be a realisation of a Gaussian process $(Z_t(\mathbf{x}))$ with:
-    mean (also named trend) $\mu_t(\mathbf{x}) = \sum_{j=1}^d b_j g_j(\mathbf{x})$ (where $g_j$ are fixed basis functions, and
5         $b_j$ are the regression coefficients of the d input variables);
-    stationary covariance function $k_t(.,.)$ (named kernel) written as $\forall \mathbf{x}, \mathbf{x}', k_t(\mathbf{x}, \mathbf{x}') =$
7         $\text{cov}(Z_t(\mathbf{x}), Z_t(\mathbf{x}'))$.

For new offshore forcing conditions $\mathbf{x}^*$, the predictive probability distribution
$Z_t(\mathbf{x}^*)| \{Z_t(\mathbf{x}^1) = z_t^1, …, Z_t(\mathbf{x}^n) = z_t^n\}$ follows a GP with mean $\hat{z}_t(\mathbf{x}^*)$ and variance $s_t^2(\mathbf{x}^*)$
defined using the universal kriging equations (e.g. Roustant et al., 2012) as follows:
$\hat{z}_t(\mathbf{x}^*) = \mathbf{g}(\mathbf{x}^*)'\hat{b} + \mathbf{c}(\mathbf{x}^*)'. \mathbf{C}^{-1}. (\mathbf{z}_t - \mathbf{G}\hat{b}),$ $\qquad\qquad$ (A1)
$s_t^2(\mathbf{x}^*) = V_S + \left(\mathbf{g}(\mathbf{x}^*)'\hat{b} - \mathbf{c}(\mathbf{x}^*)'. \mathbf{C}^{-1}. \mathbf{G}\right)'. (\mathbf{G}'. \mathbf{C}^{-1}. \mathbf{G})^{-1}. \left(\mathbf{g}(\mathbf{x}^*)'\hat{b} - \mathbf{c}(\mathbf{x}^*)'. \mathbf{C}^{-1}. \mathbf{G}\right),$ (A2)
where $\mathbf{z}_t = (Z_t(\mathbf{x}^1) = z_t^1, …, Z_t(\mathbf{x}^n) = z_t^n)$, $\mathbf{C}$ is the n×n covariance matrix between the points
$Z_t(\mathbf{x}^1),…,Z_t(\mathbf{x}^n)$ whose element is $C[i, j] = k_t(\mathbf{x}^i, \mathbf{x}^j)$; $\mathbf{c}(\mathbf{x}^*)$ is the n-dimensional vector
composed of the covariance between $Z_t(\mathbf{x}^*)$ and the points $Z_t(\mathbf{x}^1),…, Z_t(\mathbf{x}^n)$; $\mathbf{g}(\mathbf{x}^*)$ is the d-
dimensional vector of trend functions values at $\mathbf{x}^*$, $\mathbf{G} = (\mathbf{g}(\mathbf{x}^1), …, \mathbf{g}(\mathbf{x}^n))'$ is the n×d
experimental matrix, the best linear estimator $\hat{\mathbf{b}}$ of $\mathbf{b}$ is $(\mathbf{G}'\mathbf{C}^{-1}\mathbf{G})^{-1}\mathbf{G}'\mathbf{C}^{-1}\mathbf{m}_Y$, and $V_S = \sigma^2 -$
$\mathbf{c}(\mathbf{x}^*)'. \mathbf{C}^{-1}. \mathbf{c}(\mathbf{x}^*)$ by assuming $k(.,.)$ to be stationary with $\sigma^2$ a hyperparameter (named
process variance) to be estimated.





**Appendix B: Autoencoder**
This method belongs to the class of deep neural networks whose typical architecture is depicted
in Fig. B1. It consists of an input layer (left blue layer in Fig. B1), a given number of hidden
layers (light-coloured layers), and an output layer (right green layer). The process of going from
the input layer to the hidden layer is called encoding, (i.e. from the original data $y$ to latent
variables $z$), while the process of going from the hidden layer to the output layer is called
decoding (i.e. from latent variables $z$ to the variables back-transformed in the physical domain
$\hat{y}$). The central layer is named the bottleneck hidden layer and provides the latent variables $z$.
The AE architecture can be parametrised in different manners (Pawar & Attar 2019), i.e. the
number of nodes of the hidden layer, the number of nodes of the and bottleneck layer (i.e. the
number of latent variables), the type of activation function applied to each node, etc.

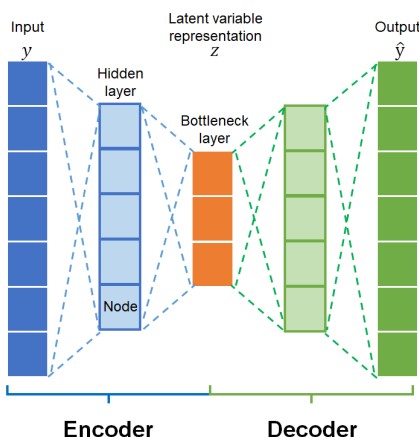

**Figure B1: Typical autoencoder architecture.**

To select the parameters of the AE architecture, we conducted the 10-fold cross validation
procedure by assuming different assumptions, namely by varying the number of nodes of the
hidden layer from 10 to 50, the number of latent variables from 3 to 5, the type of activation
function among *relu* and *sigmoid* for the hidden layer and among *linear*, *tanh* and *selu*. Figure
B2 shows the cross-validation prediction error, here measured by the root mean square error
denoted RMSE, for two sectors, Gujan-Mestras and Andernos. This indicates that having 10
nodes in the hidden layer with *relu* activation, and 3 in the bootleneck with *tanh* allows to
achieve the lowest prediction error for mesh nodes with Hmax>0 and Hmax=0 in the numerical
simulation. This result was also confirmed for the other sectors on the Arcachon lagoon.

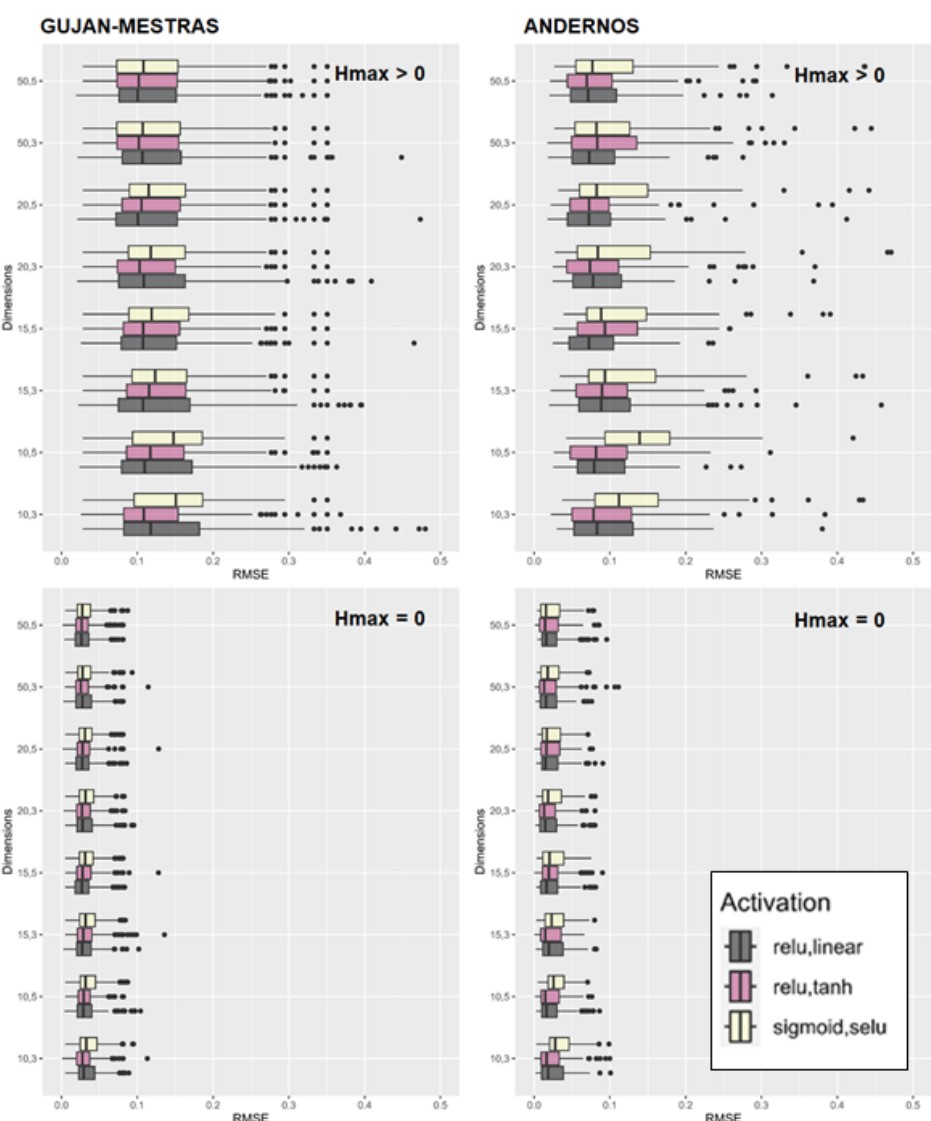

**Figure B2: Evolution of the cross-validation prediction error, here measured by the root mean square error denoted RMSE, for two sectors, Gujan-Mestras (left) and Andernos (right) for mesh nodes with Hmax>0 (top) and Hmax=0 (bottom) in the numerical simulation.**





**Appendix C: MAE for hmax=0 in the simulation for the cross validation**
The cross-validation-based MAE calculated for non-flooded mesh nodes, i.e. with Hmax=0 in
the numerical simulation, shows a clear decrease from M1 to M3. This indicates that M3
achieves the most accurate prediction of the flood spatial extent, i.e. here corresponding to the
total number of mesh nodes predicted with Hmax=0 while being non-flooded in the numerical
simulation. This suggests also that M3 enables to decrease the rate of false alarms, though we
notice that MAE remains non-zero (of the order of less than 1 cm), hence reflecting a remaining
tendency of M3 to slightly over-estimate the flood extent.

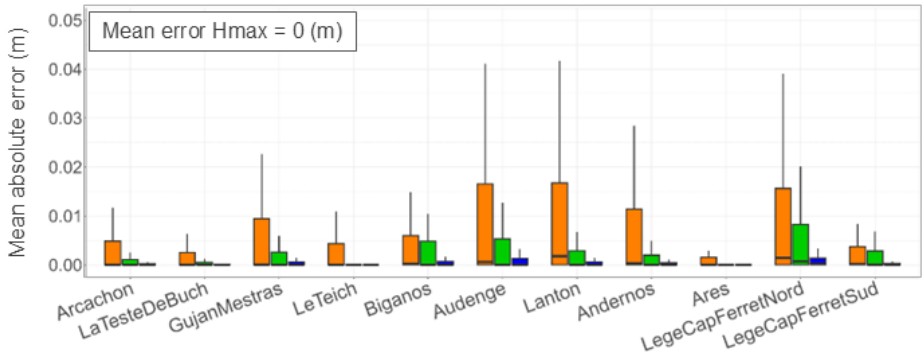

**Figure C1: Mean absolute error (MAE) calculated over the non-flooded mesh nodes (with Hmax=0 in the**
**numerical simulations) of each municipality from the cross-validation procedure.**



**1    Data availability**

All the data produced as part of the work of this work are available on request

**3    Author contribution**

JR and DI guided the development and application of the statistical methodologies. DP, AD
and DA provided the meteoceanic conditions of historical storms. AF, RP and SL completed
the marine flooding simulations. SG and SL conducted the analysis with local users on marine
flooding categories. EM performed the cross and historical validation. SL prepared the
manuscript with contributions from all co-authors.

**9    Competing interests**

The authors declare that they have no conflict of interest

**11    Acknowledgements**

This work is supported by the French National Research Agency within the ORACLES
project (ANR – 21 – CE04-0012-01).



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
