# Peer review of "INVESTIGATING METAMODELING CAPABILITY TO PREDICT SEA"

_EGUsphere, 2024_

## Referee Comment (RC1)

The paper compares three rapid coastal flood prediction methods built on a pre-computed scenario database for the Arcachon lagoon:

- M1: analog approach
- M2: Gaussian process (GP) regression for SSHmax combined with analog flood maps
- M3: GP + autoencoder (AE) for dimensionality reduction to predict Hmax maps

The connection between operational needs (CAT thresholds) and model capability is clear and relevant for early-warning applications.

Overall, the paper is scientifically solid, clearly written, and makes a timely contribution to rapid coastal flood prediction by combining physics-based simulations with data-driven methods. The methodology is sound and the results are convincing. However, some aspects, particularly the validation design, treatment of uncertainty, and clearer performance reporting across municipalities, need clarification and minor expansion to strengthen the study's rigor and operational relevance. I recommend publication after minor-to-moderate revision.

**Major Comments:**

1- The study excludes wave overtopping (Andernos seafront) and river/coincidence events in the Leyre delta, focusing only on overflow. This is fine, but in several places the text could make it clearer that results shouldn't be generalized beyond that scope. Please highlight the likely direction of bias caused by these omissions (e.g., underestimation near overtopping zones, possible CAT misranking in areas with fluvial–marine interaction). Mention these limitations again in the Discussion or Conclusion.

2- The study assumes stationary metocean conditions through the tidal cycle, supported by tests showing less than 10 cm difference in SSHmax and similar flood extents for eight events. This is useful, but the sample is small and specific to this site.

- Provide uncertainty estimates for the "<10 cm" result (e.g., confidence intervals) and, if any, show one case where dynamic effects matter, or clearly state that none were found.
- Discuss when this assumption could fail, for example, under rapidly changing winds or asymmetric surge conditions.

3- Page 14, Line 26: The validation uses only a few historical and pseudo-historical events. Please give the exact number of cases per CAT level and per municipality so readers can see how balanced the data are. Currently, only totals are given.

4- You mainly use per-node MAE and an Hmax-class confusion matrix. Consider adding flood extent metrics such as IoU/Jaccard or F1 (wet/dry) to complement these results.

5- The GP models provide predictive variances, yet the results are shown as single-value predictions. Since this is meant for early warning, please show how uncertainty affects (i) CAT probabilities and (ii) exceedance probability maps for Hmax (e.g., P[Hmax > 0.1 m]). Even one illustrative example for M2/M3 would help.

6- The tri-variate extreme analysis and clustering (50 metocean types × 10 tide levels) are well explained. A figure showing the coverage of (U, Hs, SPM) space at each tide level would reassure readers that the training data cover the full range of conditions.

7- The Klaus 2.0 and 2.6 examples are strong. If possible, include scatter plots of simulated vs. predicted Hmax (with density shading) to better illustrate spatial performance.

8- The text says M2/M3 "minimise false positives but at the expense of false alarms." Since a false alarm is a false positive, please clarify whether the models tend to over-predict (more false positives) or under-predict (more false negatives).

**Minor Comments:**

1- I see both "metamodelling" and "metamodeling" in the text. Pick one and use it throughout.
2- I recommend using "covariates" instead of "co-variables" where appropriate.
3- Page 8, Line 6: Choose either "reanalyses" (UK) or "reanalyzes" (US) and use it consistently.
4- Page 23, Table 3: Replace "Recommandation" with "Recommendation", and "prepardness" with "preparedness."
5- Page 26, Line 18: Replace "bootleneck" with "bottleneck.".
6- Define NGF early and keep units consistent in text and figures.
7- In Fig. 13 (confusion matrices), include overall accuracy and per-class F1; note which class accounts for most errors if possible.
8- When describing CAT thresholds (10 points), include the actual SSHmax ranges in a small table for quick reference. This will help reproducibility.